# LEARNING FROM OTHERS' MISTAKES: AVOIDING DATASET BIASES WITHOUT MODELING THEM

**Victor Sanh[1], Thomas Wolf[1], Yonatan Belinkov[2]\*, Alexander M. Rush[1]**
[1]Hugging Face, [2]Technion – Israel Institute of Technology
{`victor,thomas,sasha`}`@huggingface.co`;`belinkov@technion.ac.il`

## ABSTRACT

State-of-the-art natural language processing (NLP) models often learn to model dataset biases and surface form correlations instead of features that target the intended underlying task. Previous work has demonstrated effective methods to circumvent these issues when knowledge of the bias is available. We consider cases where the bias issues may not be explicitly identified, and show a method for training models that learn to ignore these problematic correlations. Our approach relies on the observation that models with limited capacity primarily learn to exploit biases in the dataset. We can leverage the errors of such limited capacity models to train a more robust model in a product of experts, thus bypassing the need to hand-craft a biased model. We show the effectiveness of this method to retain improvements in out-of-distribution settings even if no particular bias is targeted by the biased model.

## 1 INTRODUCTION

The natural language processing community has made tremendous progress in using pre-trained language models to improve predictive accuracy (Devlin et al., 2019; Raffel et al., 2019). Models have now surpassed human performance on language understanding benchmarks such as Super-GLUE (Wang et al., 2019). However, studies have shown that these results are partially driven by these models detecting superficial cues that correlate well with labels but which may not be useful for the intended underlying task (Jia & Liang, 2017; Schwartz et al., 2017). This brittleness leads to overestimating model performance on the artificially constructed tasks and poor performance in out-of-distribution or adversarial examples.

A well-studied example of this phenomenon is the natural language inference dataset MNLI (Williams et al., 2018). The generation of this dataset led to spurious surface patterns that correlate noticeably with the labels. Poliak et al. (2018) highlight that negation words ("not", "no", etc.) are often associated with the contradiction label. Gururangan et al. (2018), Poliak et al. (2018), and Tsuchiya (2018) show that a model trained solely on the hypothesis, completely ignoring the intended signal, reaches strong performance. We refer to these surface patterns as *dataset biases* since the conditional distribution of the labels given such biased features is likely to change in examples outside the training data distribution (as formalized by He et al. (2019)).

A major challenge in representation learning for NLP is to produce models that are robust to these dataset biases. Previous work (He et al., 2019; Clark et al., 2019; Mahabadi et al., 2020) has targeted removing dataset biases by explicitly factoring them out of models. These studies explicitly construct a biased model, for instance, a hypothesis-only model for NLI experiments, and use it to improve the robustness of the main model. The core idea is to encourage the main model to find a different explanation where the biased model is wrong. During training, products-of-experts ensembling (Hinton, 2002) is used to factor out the biased model.

While these works show promising results, the assumption of knowledge of the underlying dataset bias is quite restrictive. Finding dataset biases in established datasets is a costly and time-consuming process, and may require access to private details about the annotation procedure, while actively re-

---

\*Supported by the Viterbi Fellowship in the Center for Computer Engineering at the Technion

ducing surface correlations in the collection process of new datasets is challenging given the number of potential biases (Zellers et al., 2019; Sakaguchi et al., 2020).

In this work, we explore methods for learning from biased datasets which do not require such an explicit formulation of the dataset biases. We first show how a model with limited capacity, which we call a *weak learner*, trained with a standard cross-entropy loss learns to exploit biases in the dataset. We then investigate the biases on which this weak learner relies and show that they match several previously manually identified biases. Based on this observation, we leverage such limited capacity models in a product of experts ensemble to train a more robust model and evaluate our approach in various settings ranging from toy datasets up to large crowd-sourced benchmarks: controlled synthetic bias setup (He et al., 2019; Clark et al., 2019), natural language inference (McCoy et al., 2019b), extractive question answering (Jia & Liang, 2017) and fact verification Schuster et al. (2019).

Our contributions are the following: (a) we show that weak learners are prone to relying on shallow heuristics and highlight how they rediscover previously human-identified dataset biases; (b) we demonstrate that we do not need to explicitly know or model dataset biases to train more robust models that generalize better to out-of-distribution examples; (c) we discuss the design choices for weak learners and show trade-offs between higher out-of-distribution performance at the expense of the in-distribution performance.

## 2 RELATED WORK

Many studies have reported dataset biases in various settings. Examples include visual question answering (Jabri et al., 2016; Zhang et al., 2016), story completion (Schwartz et al., 2017), and reading comprehension (Kaushik & Lipton, 2018; Chen et al., 2016). Towards better evaluation methods, researchers have proposed to collect "challenge" datasets that account for surface correlations a model might adopt (Jia & Liang, 2017; McCoy et al., 2019b). Standard models without specific robust training methods often drop in performance when evaluated on these challenge sets.

While these works have focused on data collection, another approach is to develop methods allowing models to ignore dataset biases during training. Several active areas of research tackle this challenge by adversarial training (Belinkov et al., 2019a;b; Stacey et al., 2020), example forgetting (Yaghoobzadeh et al., 2019) and dynamic loss adjustment (Cadène et al., 2019). Previous work (He et al., 2019; Clark et al., 2019; Mahabadi et al., 2020) has shown the effectiveness of product of experts to train un-biased models. In our work, we show that we do not need to explicitly model biases to apply these de-biasing methods and can use a more general setup than previously presented.

Orthogonal to these evaluation and optimization efforts, data augmentation has attracted interest as a way to reduce model biases by explicitly modifying the dataset distribution (Min et al., 2020; Belinkov & Bisk, 2018), either by leveraging human knowledge about dataset biases such as swapping male and female entities (Zhao et al., 2018) or by developing dynamic data collection and benchmarking (Nie et al., 2020). Our work is mostly orthogonal to these efforts and alleviates the need for a human-in-the-loop setup which is common to such data-augmentation approaches.

Large pre-trained language models have contributed to improved out-of-distribution generalization (Hendrycks et al., 2020). However, in practice, that remains a challenge in natural language processing (Linzen, 2020; Yogatama et al., 2019) and our work aims at out-of-distribution robustness without significantly compromising in-distribution performance.

Finally, in parallel the work of Utama et al. (2020) presents a related de-biasing method leveraging the mistakes of *weakened models* without the need to explicitly model dataset biases. Our approach is different in several ways, in particular we advocate for using limited capacity weak learner while Utama et al. (2020) uses the same architecture as the robust model trained on a few thousands examples. We investigated the trade-off between learner's capacity and resulting performances as well as the resulting few-shot learning regime in the limit of a high capacity weak model.

## 3 METHOD

### 3.1 OVERVIEW

Our approach utilizes *product of experts* (Hinton, 2002) to factor dataset biases out of a learned model. We have access to a training set $(x_i, y_i)_{1 \leq i \leq N}$ where each example $x_i$ has a label $y_i$ among $K$ classes. We use two models $f_W$ (weak) and $f_M$ (main) which produce respective logits vectors $\mathbf{w}$ and $\mathbf{m} \in \mathbb{R}^K$. The product of experts ensemble of $f_W$ and $f_M$ produces logits vector $\mathbf{e}$

$$\forall 1 \leq j \leq K, e^j = w^j + m^j \tag{1}$$

Equivalently, we have $\text{softmax}(\mathbf{e}) \propto \text{softmax}(\mathbf{w}) \odot \text{softmax}(\mathbf{m})$ where $\odot$ is the element-wise multiplication.

Our training approach can be decomposed in two successive stages: (a) training the weak learner $f_W$ with a standard cross-entropy loss (CE) and (b) training a main (robust) model $f_M$ via product of experts (PoE) to learn from the errors of the weak learner. The core intuition of this method is to encourage the robust model to learn to make predictions that take into account the weak learner's mistakes.

We do not make any assumption on the biases present (or not) in the dataset and rely on letting the weak learner discover them during training. Moreover, in contrast to prior work (Mahabadi et al., 2020; He et al., 2019; Clark et al., 2019) in which the weak learner had a hand-engineered bias-specific structure, our approach does not make any specific assumption on the weak learner such as its architecture, capacity, pre-training, etc. The weak learner $f_W$ is trained with standard cross-entropy.

The final goal is producing main model $f_M$. After training, the weak model $f_W$ is frozen and used only as part of the product of experts. Since the weak model is frozen, only the main model $f_M$ receives gradient updates during training. This is similar to He et al. (2019); Clark et al. (2019) but differs from Mahabadi et al. (2020) who train both weak and main models jointly. For convenience, we refer to the cross-entropy of the prediction $\mathbf{e}$ of Equation 1 as the *PoE cross-entropy*.

### 3.2 ANALYSIS: THE ROBUST MODEL LEARNS FROM THE ERRORS OF THE WEAK LEARNER

To better explore the impact of PoE training with a weak learner, we consider the special case of binary classification with logistic regression. Here $w$ and $m$ are scalar logits and the softmax becomes a sigmoid. The loss of the product of experts for a single positive example is:

$$\mathcal{L}_{PoE,binary} = -m - w + \log\big(1 + \exp(m + w)\big) \tag{2}$$

Logit $w$ is a fixed value since the weak learner is frozen. We also define the entropy of the weak learner as $\mathcal{H}_w = -p \log(p) - (1 - p) \log(1 - p)$ where $p = \sigma(w)$ as our measure of certainty.

Different values of $w$ from the weak learner induce different gradient updates in the main model. Figure 1a shows the gradient update of the main model logit $m$. Each of the three curves corresponds to a different value of $w$ the weak model.

- **Weak Model is Certain / Incorrect**: the first case (in blue) corresponds to low values of $w$. The entropy is low and the loss of the weak model is high. The main model receives gradients even when it is classifying the point correctly ($\approx m = 5$) which encourages $m$ to compensate for the weak model's mistake.
- **Weak Model is Uncertain**: the second case (in red) corresponds to $w = 0$ which means the weak model's entropy is high (uniform probability over all classes). In this case, product of experts is equal to the main model, and the gradient is equal to the one obtained with cross-entropy.
- **Weak Model is Certain / Correct**: the third case (in green) corresponds to high values of $w$. The entropy is low and the loss of the weak model is low. In this case, $m$'s gradients are "cropped" early on and the main model receives less gradients on average. When $w$ is extremely high, $m$ receives no gradient (and the current example is simply ignored).

Put another way, the logit values for which $m$ receives gradients are shifted according the correctness and certainty of the weak model. Figure 1b shows the concentration of training examples of MNLI

(Williams et al., 2018) projected on the 2D coordinates (correctness, certainty) from a trained weak learner (described in Section 4.1). We observe that there are many examples for the 3 cases. More crucially, we verify that the group **certain / incorrect** is not empty since the examples in this group encourage the model to not rely on the dataset biases.

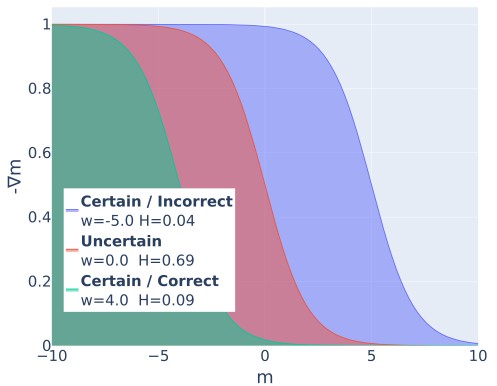

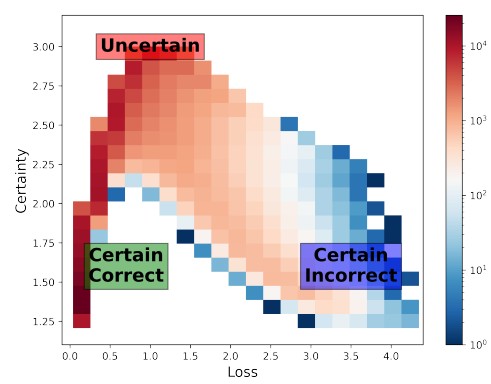

(a) Gradient update of $m$ for different values of $w$ on binary classification.

(b) 2D projection of MNLI examples from a trained weak learner. Colors indicate the concentration and are in log scale.

Figure 1: The analysis of the gradients reveals 3 regimes where the gradient is shifted by the certainty and correctness of the weak learner. These 3 regions are present in real dataset such as MNLI.

### 3.3 CONNECTION TO DISTILLATION

Our product of experts setup bears similarities with *knowledge distillation* (Hinton et al., 2015) where a student network is trained to mimic the behavior of a frozen teacher model.

In our PoE training, we encourage the main model $f_M$ (analog to the student network) to learn from the errors of the weak model $f_W$ (analog to the teacher network): instead of mimicking, it learns an orthogonal behavior when the teacher is incorrect. To recognize errors from the weak learner, we use the gold labels which alleviates the need to use pseudo-labelling or data augmentation as it is commonly done in distillation setups (Furlanello et al., 2018; Xie et al., 2020).

Similarly to Hinton et al. (2015), our final loss is a linear combination of the original cross-entropy loss (CE) and the PoE cross-entropy. We refer to this multi-loss objective as *PoE + CE*.

## 4 EXPERIMENTS

We consider several different experimental settings that explore the use of a weak learner to isolate and train against dataset biases. All the experiments are conducted on English datasets, and follow the standard setup for BERT training. Our main model is BERT-base (Devlin et al., 2019) with 110M parameters. Except when indicated otherwise, our weak learner is a significantly smaller pre-trained masked language model known as TinyBERT (Turc et al., 2019) with 4M parameters (2 layers, hidden size of 128). The weak learner is fine-tuned on exactly the same data as our main model. For instance, when trained on MNLI, it gets a 67% accuracy on the matched development set (compared to 84% for BERT-base).

Part of our discussion relies on natural language inference, which has been widely studied in this area. The classification task is to determine whether a *hypothesis* statement is true (entailment), false (contradiction) or undetermined (neural) given a *premise* statement. MNLI (Williams et al., 2018) is the canonical large-scale English dataset to study this problem with 433K labeled examples. For evaluation, it features matched sets (examples from domains encountered in training) and mismatched sets (domains not-seen during training).

Experiments first examine qualitatively the spurious correlations picked up by the method. We then verify the validity of the method on a synthetic experimental setup. Finally, we verify the impact of our method by evaluating robust models on several out-of-distribution sets and discuss the choice of the weak learner.

## 4.1 WEAK LEARNERS REDISCOVER PREVIOUSLY REPORTED DATASET BIASES

Most approaches for circumventing dataset bias require modeling the bias explicitly, for example using a model limited to only the hypothesis in NLI (Gururangan et al., 2018). These approaches are effective, but require isolating specific biases present in a dataset. Since this process is costly, time consuming and error-prone, it is unrealistic to expect such analysis for all new datasets. On the contrary, we hypothesize that weak learners might operate like *rapid surface learners* (Zellers et al., 2019), picking up on dataset biases without specific signal or input curation and being rather certain of their biased errors (high certainty on the biased prediction errors).

Table 1: Breakdown of the 1,000 top **certain / incorrect** training examples.

| Category | (%) |
|---|---|
| Predicted *Contradiction* | 46 |
| Neg. in the hyp. | 43 |
| Predicted *Entailment* | 51 |
| High word overlap prem./hyp. | 43 |
| Predicted *Neutral* | 3 |

We first investigate whether our weak learner re-discovers two well-known dataset biases reported on NLI benchmarks: (a) the presence of negative word in the hypothesis is highly correlated with the *contradiction* label (Poliak et al., 2018; Gururangan et al., 2018), (b) high word overlap between the premise and the hypothesis is highly correlated with the *entailment* label (McCoy et al., 2019b).

To this aim, we fine-tune a weak learner on MNLI (Williams et al., 2018). Hyper-parameters can be found in Appendix A.1. We extract and manually categorize 1,000 training examples wrongly predicted by the weak learner (with a **high loss** and a **high certainty**). Table 1 breaks them down per category. Half of these incorrect examples are wrongly predicted as *Contradiction* and almost all of these contain a negation[1] in the hypothesis. Another half of the examples are incorrectly predicted as *Entailment*, a majority of these presenting a high lexical overlap between the premise and the hypothesis (5 or more words in common). The weak learner thus appears to predict with high-certainty a *Contradiction* label whenever the hypothesis contains a negative word and with high-certainty an *Entailment* label whenever there is a strong lexical overlap between premise/hypothesis. Table 6 in Appendix A.3 presents qualitative examples of dataset biases identified by the fine-tuned weak learner.

This analysis is based on a set of biases referenced in the literature and does not exclude the possibility of other biases being detected by the weak learner. For instance, during this investigation we notice that the presence of "negative sentiment" words (for instance: *dull, boring*) in the hypothesis appears to be often indicative of a *Contradiction* prediction. We leave further investigation on such behaviors to future work.

## 4.2 SYNTHETIC EXPERIMENT: CHEATING FEATURE

We consider a controlled synthetic experiment described in He et al. (2019); Clark et al. (2019) that simulates bias. We modify 20,000 MNLI training examples by injecting a *cheating feature* which encodes an example's label with probability $p_{cheat}$ and a random label selected among the two incorrect labels otherwise. For simplicity, we consider the first 20,000 examples. On the evaluation sets, the cheating feature is random and does not convey any useful information. In the present experiment, the cheating feature takes the form of a prefix added to the hypothesis ("0" for *Contradiction*, "1" for *Entailment*, "2" for *Neutral*). We train the weak and main models on these 20,000 examples and evaluate their accuracy on the matched development set.[2] We expect a biased model to rely mostly on the cheating feature thereby leading to poor evaluation performance.

Figure 2 shows the results. As the proportion of examples containing the bias increases, the evaluation accuracy of the weak learner quickly decreases to reach 0% when $p_{cheat} = 0.9$. The weak

---

[1] We use the following list of negation words: *no, not, none, nothing, never, aren't, isn't, weren't, neither, don't, didn't, doesn't, cannot, hasn't, won't*.

[2] We observe similar trends on the mismatched development set.

learner detects the cheating feature during training and is mainly relying on the synthetic bias which is not directly indicative of the gold label.

Both He et al. (2019) and Clark et al. (2019) protect against the reliance on this cheating feature by ensembling the main model with a biased model that only uses the hypothesis (or its first token). We instead train the main model in the product of experts setting, relying on the weak learner to identify the bias. Figure 2 shows that when a majority of the training examples contain the bias ($p_{cheat} \geq 0.6$), the performance of the model trained with cross-entropy drops faster than the one trained in PoE. PoE training leads to a more robust model by encouraging it to learn from the mistakes of the weak learner. As $p_{cheat}$ comes close to 1, the model's training enters a "few-shot regime" where there are very few incorrectly predicted biased examples to learn from (examples where following the biased heuristic lead to a wrong answer) and the performance of the model trained with PoE drops as well.

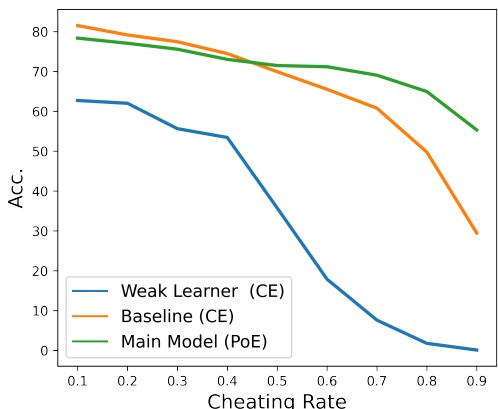

Figure 2: Accuracy on MNLI matched development set for models with a cheating feature. The model trained with PoE (*Main Model*) is less sensitive to this synthetic bias.

### 4.3 ADVERSARIAL DATASETS: NLI AND QA

**NLI** The HANS adversarial dataset (McCoy et al., 2019b) was constructed by writing templates to generate examples with a high premise/hypothesis word overlap to attack models that rely on this bias. In one template the word overlap generates entailed premise/hypothesis pairs (*heuristic-entailed examples*), whereas in another the examples contradict the heuristic (*non-heuristic-entailed*). The dataset contains 30K evaluation examples equally split between both.

Table 2 shows that the weak learner exhibits medium performance on the in-distribution sets (MNLI) and that on out-of-distribution evaluation (HANS), it relies heavily on the word overlap heuristic. Product of experts training is effective at reducing the reliance on biases and leads to significant gains on the heuristic-non-entailed examples when compared to a model trained with standard cross-entropy leading to an improvement of +24%.

The small degradation on in-distribution data is likely because product of experts training does not specialize for in-distribution performance but focuses on the weak model errors (He et al., 2019). The linear combination of the original cross-entropy loss and the product of experts loss (PoE + CE) aims at counteracting this effect. This multi-loss objective trades off out-of-distribution generalization for in-distribution accuracy. A similar trade-off between accuracy and robustness has been reported in adversarial training (Zhang et al., 2019; Tsipras et al., 2019). In Appendix A.6, we detail the influence of this multi-loss objective.

We also evaluate our method on MNLI's hard test set (Gururangan et al., 2018) which is expected to be less biased than MNLI's standard split. These examples are selected such that a hypothesis-only model cannot predict the label accurately. Table 2 shows the results of this experiment. Our method surpasses the performance of a PoE model trained with a hypothesis-only biased learner. Results on the mismatched set are given in Appendix A.4.

**QA** Question answering models often rely on heuristics such as type and keyword-matching (Weissenborn et al., 2017) that can do well on benchmarks like SQuAD (Rajpurkar et al., 2016). We evaluate on the Adversarial SQuAD dataset (Jia & Liang, 2017) built by appending distractor sentences to the passages in the original SQuAD. Distractors are constructed such that they look like a plausible answer to the question while not changing the correct answer or misleading humans.

Results on SQuAD v1.1 and Adversarial SQuAD are listed in Table 3. The weak learner alone has low performance both on in-distribution and adversarial sets. PoE training improves the adversarial performance (+1% on AddSent) while sacrificing some in-distribution performance. A multi-loss

Table 2: MNLI matched dev accuracies, HANS accuracies and MNLI matched hard test set. We repo the All numbers are averaged on 6 runs (with standard deviations). Detailed results for HANS are given Appendix A.4. Reported results are indicated with *. ♣ Utama et al. (2020) is a concurrent work where they use a BERT-base fine-tuned on 2000 random examples from MNLI as a weak learner and "PoE + An." refers to the annealing mechanism proposed by the authors. *I* and *O* are respectively referring to in-distribution and out-of-distribution sets.

| | Loss | MNLI (*I*) | HANS | | Hard (*O*) |
| | | | Ent (*I*) | Non-Ent (*O*) | |
|---|---|---|---|---|---|
| Clark et al. (2019)* | PoE | 82.97 | 64.67 | 71.16 | - |
| Mahabadi et al. (2020)* | PoE | 84.19 | 95.99 | 33.30 | 76.81 |
| Utama et al. (2020)*♣ | PoE | 80.70 | 86.13 | 55.20 | - |
| Utama et al. (2020)*♣ | PoE + An. | 81.90 | 88.40 | 47.13 | - |
| BERT-base | CE | **84.52**±0.27 | 98.12±0.62 | 26.74±6.15 | 76.96±0.38 |
| TinyBERT - Weak | CE | 66.93±0.12 | **99.80**±0.09 | 0.44±0.26 | 46.65±0.48 |
| BERT-base - Main | PoE | 81.35±0.40 | 81.13±8.1 | **56.41**±5.91 | 76.54±0.56 |
| BERT-base - Main | PoE + CE | 83.32±0.24 | 94.51±0.82 | 41.35±8.25 | **77.63**±0.49 |

| | Loss | SQuAD (*I*) | Adversarial QA (*O*) | |
| | | | AddSent | AddOneSent |
|---|---|---|---|---|
| Clark et al. (2019)* | CE | 80.61 | 42.54 | 53.91 |
| BiDAF | PoE | 78.63 | 57.64 | 57.17 |
| BERT-base | CE | **88.68** | 53.98 | 58.84 |
| TinyBERT - Weak | CE | 41.08 | 16.02 | 18.63 |
| BERT-base - Main | PoE | 83.11 | 54.92 | 58.44 |
| BERT-base - Main | PoE + CE | 86.49 | **56.80** | **61.04** |

Table 3: F1 Scores on SQuAD and Adversarial QA. The AddOneSent set is model agnostic while we use the AddSent set obtained using an ensemble of BiDAF models (Seo et al., 2017). * are reported results. *I* and *O* are respectively referring to in-distribution and out-of-distribution sets.

optimization closes the gap and even boosts adversarial robustness (+3% on AddSent and +2% on AddOneSent). In contrast to our experiments on MNLI/HANS, multi-loss training thus leads here to better performance on out-of-distribution as well. We hypothesize that in this dataset, the weak learner picks up more useful information and removing it entirely might be non-optimal. Multi-loss in this case allows us to strike a balance between learning from, or removing, the weak learner.

## 5 ANALYSIS

### 5.1 REDUCING BIAS: CORRELATION ANALYSIS

To investigate the behavior of the ensemble of the weak and main learner, we compute the Pearson correlation between the element-wise loss of the weak (biased) learner and the loss of the trained models following Mahabadi et al. (2020). A correlation of 1 indicates a linear relation between the per-example losses (the two learners make the same mistakes), and 0 indicates the absence of linear correlation (models' mistakes are uncorrelated). Figure 3 shows that models trained with a linear combination of the PoE cross-entropy and the standard cross-entropy have a higher correlation than when trained solely with PoE. This confirms that PoE training is effective at reducing biases uncovered by the weak learner and re-emphasizes that adding standard cross-entropy leads to a trade-off between the two.

### 5.2 HOW WEAK DO THE WEAK LEARNERS NEED TO BE?

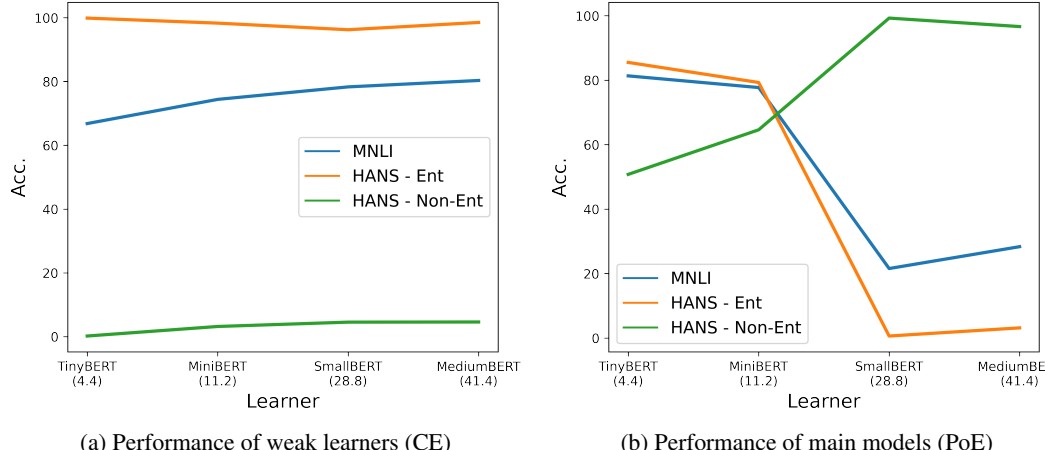

(a) Performance of weak learners (CE)  (b) Performance of main models (PoE)

Figure 4: Weaker learners assure a good balance between out-of-distribution and in-distribution while stronger learners encourage out-of-distribution generalization at the expense of in-distribution performance. We indicate the number of parameters in parenthesis (in millions).

We consider parameter size as a measure of the capacity or "weakness" of the weak learner. We fine-tune different sizes of BERT (Turc et al., 2019) ranging from 4.4 to 41.4 million parameters and use these as weak models in a PoE setting. Figure 4b shows the accuracies on MNLI and HANS of the weak learners and main models trained with various weak learners.

Varying the capacity of the weak models affects both in-distribution and out-of-distribution performance. Out-of-distribution performance of the main model increases as the weak model becomes stronger (more parameters) up to a certain point while in-distribution performances drop slightly at first and then more strongly. When trained jointly with the larger MediumBERT weak learner (41.4 million parameters), the main model gets 97% accuracy on HANS's heuristic-non-entailed set but a very low accuracy on the in-distribution examples (28% on MNLI and 3% on the heuristic-entailed examples).

As a weak model grows in capacity, it becomes a better learner. The average loss decreases and the model becomes more confident in its predictions. As a result, the group

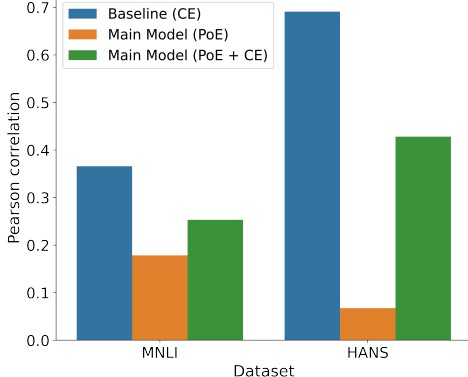

Figure 3: Pearson correlation between the losses (on evaluation sets) of the biased model and different training methods. The PoE training is effective at reducing the correlation with the biased model.

**certain / correct** becomes more populated and the main model receives on average a smaller gradient magnitude per input. On the contrary, the **certain / incorrect** group (which generally align with out-of-distribution samples and induce higher magnitude gradient updates, encouraging generalization at the expense of in-distribution performance) becomes less populated. These results corroborate and complement insights from Yaghoobzadeh et al. (2019). This is also reminiscent of findings from Vodrahalli et al. (2018) and Shrivastava et al. (2016): not all training samples contribute equally towards learning and in some cases, a carefully selected subset of the training set is sufficient to match (or surpass) the performance on the whole set.

Table 4: Accuracies on MNLI (matched) and HANS when trained on a subset MNLI. Training examples are selected from the whole MNLI training set adversarially as detailed in the text. We use BERT-base as the main model and TinyBERT as the weak leaner. *I* and *O* are respectively referring to in-distribution and out-of-distribution sets.

| Loss | Training set | MNLI (*I*) | HANS | |
| | | | Ent (*I*) | Non-Ent (*O*) |
|---|---|---|---|---|
| CE | MNLI minus negation/overlap | 60.59 | **91.50** | 24.19 |
| PoE | MNLI minus negation/overlap | **64.63** | 84.85 | **50.96** |

### 5.3 DE-BIASING IS STILL EFFECTIVE WHEN DATASET BIASES ARE UNKNOWN OR HARD TO DETECT

While it is difficult to enumerate all sources of bias, we focus in this work on superficial cues that correlate with the label in the training set but do not transfer. These superficial cues correlate with what can be captured by a weak model. For instance, Conneau et al. (2018) suggest that word presence can be detected with very shallow networks (linear classifier on top of FastText bag of words) as they show very high accuracy for Word Content, the probing task of detecting which of the 1'000 target words is present in a given sentence.

To verify that a weak model is still effective with unknown or hard to detect biases, we consider an example where the bias is only present in a small portion of the training. We remove from the MNLI training set all the examples (192K) that exhibit one of the two biases detailed in Section 4.1: high word overlap between premise and hypothesis with *entailment* label; and negation in the hypothesis with *contradiction* label. We are left with 268K training examples.

We apply our de-biasing method with these examples as our training set. For comparison, we train a main model with standard cross-entropy on the same subset of selected examples. Our results are shown in Table 4 and confirm on HANS that our de-biasing method is still effective even when the bias is hard to detect. Note that the accuracies on MNLI can not be directly compared to results in Table 2: the class imbalance in the selected subset of examples lead to a harder optimization problem explaining the difference of performance.

We present complementary analyses in Appendix. To further show the effectiveness of our method, we included in Appendix A.2 an additional experiment on facts verification (Thorne et al., 2018; Schuster et al., 2019). In Appendix A.5, we evaluate the ability of our method to generalize to other domains that do not share the same annotation artifacts. We highlight the trade-off between in-distribution performance and out-of-distribution robustness by quantifying the influence of the multi-loss objective in Appendix A.6 and draw a connection between our 3 groups of examples and recently introduced *Data Maps* (Swayamdipta et al., 2020).

## 6 CONCLUSION

We have presented an effective method for training models robust to dataset biases. Leveraging a weak learner with limited capacity and a modified product of experts training setup, we show that dataset biases do not need to be explicitly known or modeled to be able to train models that can generalize significantly better to out-of-distribution examples. We discuss the design choices for such weak learner and investigate how using higher-capacity learners leads to higher out-of-distribution performance and a trade-off with in-distribution performance. We believe that such approaches capable of automatically identifying and mitigating datasets bias will be essential tools for future bias-discovery and mitigation techniques.

## ACKNOWLEDGEMENTS

This research was supported by the ISRAEL SCIENCE FOUNDATION (grant No. 448/20).

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

# A APPENDIX

## A.1 EXPERIMENTAL SETUP AND FINE-TUNING HYPER-PARAMETERS

Our code is based on the Hugging Face Transformers library (Wolf et al., 2019). All of our experiments are conducted on single 16GB V100 using half-precision training for speed.

**NLI** We fine-tuned a pre-trained TinyBERT (Turc et al., 2019) as our weak learner. We use the following hyper-parameters: 3 epochs of training with a learning rate of $3e-5$, and a batch size of 32. The learning rate is linearly increased for 2000 warming steps and linearly decreased to 0 afterward. We use an Adam optimizer $\beta = (0.9, 0.999), \epsilon = 1e-8$ and add a weight decay of 0.1. Our robust model is `BERT-base-uncased` and uses the same hyper-parameters. When we train a robust model with a multi-loss objective, we give the standard CE a weight of 0.3 and the PoE cross-entropy a weight of 1.0. Because of the high variance on HANS (McCoy et al., 2019a), we average numbers on 6 runs with different seeds.

**SQuAD** We fine-tuned a pre-trained TinyBERT (Turc et al., 2019) as our weak learner. We use the following hyper-parameters: 3 epochs of training with a learning rate of $3e-5$, and a batch size of 16. The learning rate is linearly increased for 1500 warming steps and linearly decreased to 0 afterward. We use an Adam optimizer $\beta = (0.9, 0.999), \epsilon = 1e-8$ and add a weight decay of 0.1. Our robust model is `BERT-base-uncased` and uses the same hyper-parameters. When we train a robust model with a multi-loss objective, we give the standard CE a weight of 0.3 and the PoE cross-entropy a weight of 1.0.

**Fact verification** We fine-tuned a pre-trained TinyBERT (Turc et al., 2019) as our weak learner. We use the following hyper-parameters: 3 epochs of training with a learning rate of $2e-5$, and a batch size of 32. The learning rate is linearly increased for 1000 warming steps and linearly decreased to 0 afterward. We use an Adam optimizer $\beta = (0.9, 0.999), \epsilon = 1e-8$ and add a weight decay of 0.1. Our robust model is `BERT-base-uncased` and uses the same hyper-parameters. When we train a robust model with a multi-loss objective, we give the standard CE a weight of 0.3 and the PoE cross-entropy a weight of 1.0. We average numbers on 6 runs with different seeds.

## A.2 ADDITIONAL EXPERIMENT: FACT VERIFICATION

Following Mahabadi et al. (2020), we also experiment on a fact verification dataset. The FEVER dataset (Thorne et al., 2018) contains claim-evidences pairs generated from Wikipedia. Schuster et al. (2019) collected a new evaluation set for the FEVER dataset to avoid the biases observed in the claims of the benchmark. The authors symmetrically augment the claim-evidences pairs of the FEVER evaluation to balance the detected artifacts such that solely relying on statistical cues in claims would lead to a random guess. The collected dataset is challenging, and the performance of the models relying on biases evaluated on this dataset drops significantly.

Table 5: Accuracies on the FEVER dev set (Thorne et al., 2018) and symmetric hard test (Schuster et al., 2019).

|  | Loss | Dev | Symmetic test |
|---|---|---|---|
| Schuster et al. (2019)* | PoE | **85.85** | 57.46 |
| Mahabadi et al. (2020)* | CE | 85.99 | 56.49 |
| Mahabadi et al. (2020)* | PoE | 84.46 | **66.25** |
| BERT-base | CE | 85.61±0.3 | 55.13±1.5 |
| TinyBERT - Weak | CE | 69.43±0.2 | 43.10±0.2 |
| BERT-base - Main | PoE | 81.97±0.5 | **59.95**±3.3 |
| BERT-base - Main | PoE + CE | **85.29**±0.6 | 57.86±1.4 |

Our results are in Table 5. Our method is again effective at removing potential biases present in the training set and shows strong improvements on the symmetric test set.

### A.3 Some examples of dataset biases detected by the weak learner

In Table 6, we show a few qualitative examples of dataset biases detected by a fine-tuned weak learner.

Table 6: Weak learners are able to detect previously reported dataset biases without explicitly modeling them.

| Category | Gold / Predicted Label | Premise | Hypothesis |
|---|---|---|---|
| Negation in Hypothesis | Entailment / Contradiction | What explains the stunning logical inconsistencies and misrepresentations in this book? | Nothing can explain the stunning logical inconsistencies in this book. |
| | Entailment / Contradiction | There were many things that disturbed Jon as he stood in silence and observed the scout. | Jon didn't say anything. |
| High word overlap | Neutral / Entailment | Conservatives concede that some safety net may be necessary. | Democrats concede that some safety net may be necessary. |
| | Contradiction / Entailment | New Kingston is the modern commercial center of the capital, but it boasts few attractions for visitors. | New Kingston is a modern commercial center that has many attractions for tourists. |

### A.4 Detailed results on HANS and MNLI mismatched

We report the detailed results per heuristics on HANS in Table 7 and the results on the mismatched hard test set of MNLI in Table 8.

Table 7: HANS results per heuristic. All numbers are an average on 6 runs (with standard deviations). *I* and *O* are respectively referring to in-distribution and out-of-distribution sets.

| | Loss | Ent (*I*) | | | Non-Ent (*O*) | | |
|---|---|---|---|---|---|---|---|
| | | Lex | Subseq | Const | Lexical | Subseq | Const |
| BERT-base | CE | 99.50±0.83 | 99.20±1.16 | 99.36±0.32 | 52.43±16.45 | 10.03±4.01 | 17.76±1.65 |
| TinyBERT - Weak | CE | **100.0**±0.00 | **100.0**±0.00 | **99.40**±0.28 | 0.00±0.00 | 0.00±0.00 | 1.33±0.77 |
| BERT-base - Main | POE | 71.89±11.39 | 85.92±9.38 | 85.56±4.40 | **74.60**±10.53 | **35.70**±8.42 | **53.53**±5.60 |
| BERT-base - Main | PoE + CE | 90.17±1.94 | 97.02±1.38 | 96.32±0.58 | 66.00±20.19 | 17.97±5.53 | 40.07±4.47 |

Table 8: Accuracies on the MNLI dev mismatched and HARD test mismatched set. *I* and *O* are respectively referring to in-distribution and out-of-distribution sets.

| | Loss | MNLI (*I*) | HARD (*O*) |
|---|---|---|---|
| Mahabadi et al. (2020)* | PoE | 83.47 | 76.83 |
| BERT-base | CE | **84.87**±0.18 | **76.71**±0.48 |
| TinyBERT - Weak | CE | 68.24±0.30 | 48.53±0.49 |
| BERT-base - Main | PoE | 81.18±0.65 | 75.60±0.70 |
| BERT-base - Main | PoE + CE | 83.54±0.34 | 76.39±0.62 |

### A.5 NLI: transfer experiments

As highlighted in Section 4.3, our method is effective at improving robustness to adversarial settings that specifically target dataset biases. We further evaluate how well our method improves

generalization to domains that do not share the same annotation artifacts. Mahabadi et al. (2020) highlighted that product of experts is effective at improving generalization to other NLI benchmarks when trained on SNLI (Bowman et al., 2015). We follow the same setup: notably, we perform a sweep on the weight of the cross-entropy in our multi-loss objective and perform model selection on the development set of each dataset. We evaluate on SciTail (Khot et al., 2018), GLUE benchmark's diagnostic test (Wang et al., 2018), AddOneRTE (AddOne) (Pavlick & Callison-Burch, 2016), Definite Pronoun Resolution (DPR) (Rahman & Ng, 2012), FrameNet+ (FN+) (Pavlick et al., 2015) and Semantic Proto-Roles (SPR) (Reisinger et al., 2015). We also evaluate on the hard SNLI test set (Gururangan et al., 2018), which is a set where a hypothesis-only model cannot solve easily.

Table 9 shows the results. Without explicitly modeling the bias in the dataset, our method matches or surpasses the generalization performance previously reported, at the exception of GLUE's diagnostic dataset and SNLI's hard test set. Moreover, we notice that the multi-loss objective sometimes leads to a stronger performance, which suggests that in some cases, it can be sub-optimal to completely remove the information picked up by the weak learner. We hypothesize that the multi-loss objective balances the emphasis on domain-specific features (favoring in-distribution performance) and their removal through de-biasing (benefiting domain transfer performance). This might explain why we do not observe improvements on SNLI's hard test set and GLUE's diagnostic set in the PoE setting.

Table 9: Transfer accuracies on NLI benchmarks. We train BERT-base on SNLI and tested on the target test set. * are reported results.

| | Mahabadi et al. (2020)* | | Ours | | |
| Data | CE | PoE | CE | PoE | PoE + CE |
| --- | --- | --- | --- | --- | --- |
| AddOne | 87.34 | 87.86 | 86.82 | 85.53 | **87.34** |
| DPR | 49.50 | 50.14 | 50.29 | **50.40** | 50.34 |
| SPR | 59.85 | 62.45 | 58.27 | **62.82** | 60.50 |
| FN+ | 53.16 | 53.51 | 54.35 | 54.71 | **54.90** |
| SCITAIL | 67.64 | 71.40 | 69.71 | **74.13** | 73.57 |
| GLUE | 54.08 | 54.71 | **55.34** | 49.28 | 53.26 |
| SNLI Hard | 80.53 | 82.15 | 81.17 | 78.10 | **81.99** |

## A.6 Influence of multi-loss objective

Our best performing setup features a linear combination of the PoE cross-entropy and the standard cross-entropy. We fix the weight of the PoE cross-entropy to 1. and modulate the linear coefficient $\alpha$ of the standard cross-entropy. Figure 5 shows the influence of this multi-loss objective. As the weight of the standard cross-entropy increases, the in-distribution performance increases while the out-of-distribution performance decreases. This effect is particularly noticeable on MNLI/HANS (see Figure 5a). Surprisingly, this trade-off is less pronounced on SQuAD/Adversarial SQuAD: the F1 development score increases from 85.43 for $\alpha = 0.1$ to 88.14 for $\alpha = 1.9$ while decreasing from 56.67 to 55.06 on AddSent.

Our multi-loss objective is similar to the annealing mechanism proposed in (Utama et al., 2020). In fact, as the annealing coefficient decreases, the modified probability distribution of the weak model converges to the uniform distribution. As seen in Section 3.2, when the distribution of the weak model is close to the uniform distribution (high-uncertainty), the gradient of the loss for PoE is equivalent to the gradient of the main model trained without PoE (i.e. the standard cross-entropy). In this work, we consider a straight-forward setup where we linearly combine the two losses throughout the training with fixed coefficients.

## A.7 Connection to *Data Maps* Swayamdipta et al. (2020)

We hypothesize that our three identified groups (**certain / incorrect**, **certain / correct**, and **uncertain**) overlap with the regions identified by data cartographies (Swayamdipta et al., 2020). The authors project each training example onto 2D coordinates: confidence, variability. The first one is the mean of the gold label probabilities predicted for each example across training epochs. The second one is the standard deviation. Confidence is closely related to the loss (intuitively, a high-

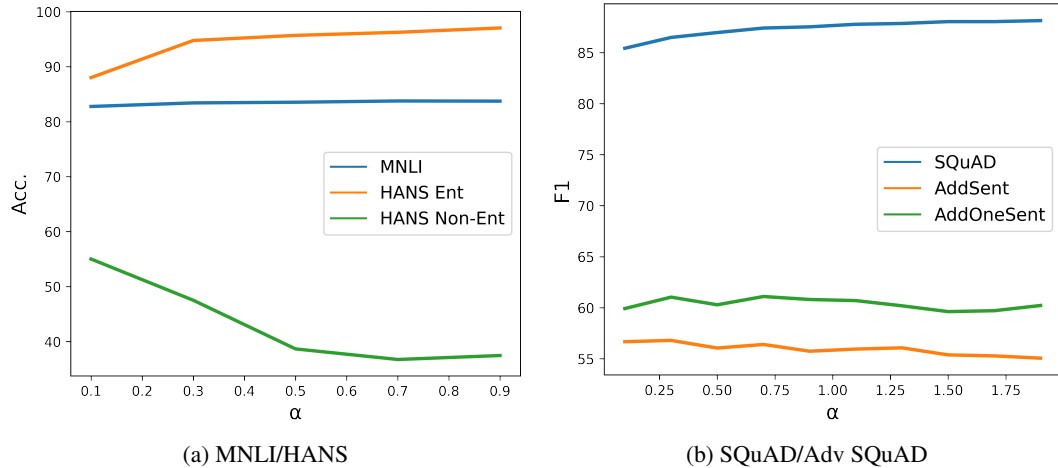

(a) MNLI/HANS                                    (b) SQuAD/Adv SQuAD

Figure 5: The multi-loss objective controls a trade-off between the in-distribution performance and out-of-distribution robustness.

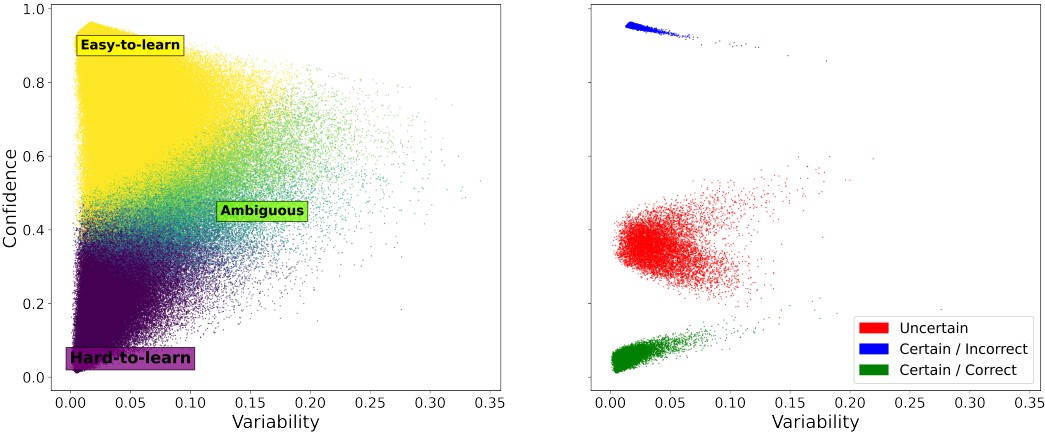

Figure 6: Our groups are included in the regions identified in *Data Maps*.

confidence example is "easier" to predict). Variability is connected to the uncertainty (the probability of the true class for high-variability examples fluctuates during the training underlying the model's indecisiveness).

Our most interesting group (**certain / incorrect**) bears similarity with the *ambiguous region* (the model is indecisive about these instances frequently changing its prediction across training epochs) and the *hard-to-learn region* (which contains a significant proportion of mislabeled examples). The authors observe that the examples in these 2 regions play an important role in out-of-distribution generalization. Our findings go in the same direction as the weak model encourages the robust model to pay a closer look at these **certain / incorrect** examples during training.

To verify this claim, we follow their procedure and log the training dynamics of our weak learner trained on MNLI. Figure 6 (left) shows the *Data Map* obtained with our weak learner. For each of our 3 groups, we select the 10,000 examples that emphasized the most the characteristic of the group. For instance, for **uncertain**, we take the 10,000 examples with the highest entropy. In Figure 6 (right) we plot these 30,000 examples onto the *Data Map*. We observe that our **certain / correct** examples are in the *easy-to-learn* region, our **certain / incorrect** examples are in the *hard-to-learn* region and our **uncertain** examples are mostly in the *ambiguous* region.

Conversely, we verify that the examples in the *ambiguous* region are mostly in our **uncertain** group, the examples from the *hard-to-learn* are mostly in **uncertain** and **certain / incorrect**, and the examples from the *easy-to-learn* are mainly in **certain / correct** and **uncertain**.

