# OpenReview forum: "Learning from others' mistakes: Avoiding dataset biases without modeling them"
_ICLR.cc/2021/Conference — ICLR 2021 Poster_

### Official Review · AnonReviewer1 · 2020-10-18

**Rating:** 2
**Confidence:** 5

**Review:**

Paper summary:
The authors argue that they have proposed a method to train robust models to biases without having prior knowledge of the biases. They argue also to provide analysis on how weak learner capacity impacts the in-domain/out-of-domain performance.

Reasons to reject:
1) The authors argue they have shown the model with limited capacity capture biases. However, this has been shown already in [1] in 2019 and therefore is not a contribution of the authors.
2) The main method proposed in this paper, is exactly the same method proposed in [2]. Please note that [2] was already available in early July 2020, and on top of existing work, the paper does not provide other contributions.
3) About the third argued contribution on showing how the performance of the debiasing method change based on the capacity of weak learners, in [1], the authors included the discussion between the choice of weak learners on their impact. Though the method in [1] is different, the discussion in that paper still would apply here as well.  Please refer to table 1-3 and Figure 1 in [1].

Given the points above, and since the main method in the paper is proposed in [2], the paper does not provide enough contributions to be suitable for the ICLR venue.

[1] Robust Natural Language Inference Models with Example Forgetting, Yaghoobzadeh et al, https://arxiv.org/pdf/1911.03861.pdf, 2019
[2] Towards Debiasing NLU Models from Unknown Biases, Utama et al, 13 July 2020, https://openreview.net/forum?id=UHpxm2K-jHE, EMNLP 2020

---

> ### Comment · Area_Chair1 · 2020-10-25
> **Clarification on ICLR policy**
>
> Thank you for your review! I wanted to clarify that ICLR policy is that not citing/comparing with unpublished work, particularly recent work, may be excused. See https://iclr.cc/Conferences/2021/ReviewerGuide
>
> Q: Are authors expected to cite and compare with very recent work? What about non peer-reviewed (e.g., ArXiv) papers?
>
> A: We consider papers contemporaneous if they are published within the last two months. That means, since our full paper deadline is Oct 2, if a paper was published on or after Aug 2, 2020, authors are not required to compare their own work to that paper. Authors are encouraged to cite and discuss all relevant papers, but they may be excused for not knowing about papers not published in peer-reviewed conference proceedings or journals.
>
>
>
> Since [2] was submitted to EMNLP by June 3, and notifications of acceptance were Sept. 14, this work can be considered contemporaneous. Of course, it is helpful to point authors to this work and ask them to cite it, but please consider reviewing the paper on its own merits.

---

> ### Author Response · Authors · 2020-11-24
> **Response to Reviewer1**
>
>
> We want to sincerely thank you for your comments. We are encouraged to see that this area of research is very active and that multiple concurrent works are proposed independently.
>
> We want to address to your comments:
>
> 1. **The authors argue they have shown the model with limited capacity capture biases. However, this has been shown already**
>
> In [1], the authors also look at models with low capacity to discover examples that support the heuristics. However, they use a different definition of "hard examples": forgotten examples. From [1], "an example is forgotten if it goes from being correctly to incorrectly classified (because of multiple gradient updates performed on other examples)." It is similar to the definition used in Data Cartography [3] for which we show in Appendix A6 that the groups they identified strongly overlap with our groups whose definition of hardness is based on the pair (final loss, final uncertainty). Our results show that we do not need to track the whole fine-tuning of our low-capacity weak learner and yet can produce similar improvements.
>
> From a practical point of view, we can imagine a setup where one could use a publicly released fine-tuned checkpoint (such as the ones found on huggingface.co/models) and use it as (an already trained) weak learner.
>
> 2. **The main method proposed in this paper, is exactly the same method proposed in [2].**
>
> We can now see that [2] was available on OpenReview mid-July 2020 as an anonymous pre-print. We became aware of it when it was posted to arxiv September 25th, 2020 (it was finally published in EMNLP2020’s proceedings on November 9th, 2020). Upon becoming aware of this work as preparing for this submission (October 2nd, 2020), we mentioned it explicitly in our manuscript and added its results. We believe this was following proper research protocol, and think that our work should be evaluated independently from the concurrent work of [2].
>
> We also want to highlight a core difference with this work. In [2], the authors use the same high-capacity model (BERT-base) for both the weak model and the main model. They “control” the weak learner by only presenting a tiny fraction of the data (for instance, 2’000 examples for MNLI randomly sampled among the 392’000 training examples). In contrast, we “control” the weak learner by limiting its capacity (number of parameters) but fine-tune it on the whole training dataset. We argue that the method used to “control” the weak learner in [2] can have drawbacks. Namely, it fails to leverage a significant proportion of the signal present in the training set.
>
> To fairly compare the two setups, we design an extreme experiment: we adversarially sample the 2K examples fed to the weak learner by training a hypothesis-only classifier and selecting the examples this classifier can’t correctly classify. Intuitively, these examples are harder to classify because the bias is not (or less) present.
> We observe that the generalization decreases compared to a randomly sampled set of 2k examples at no cost of performance on in-domain inputs. This suggests that there are unfortunate small subsamples of (hard) examples that provide less debiasing ability to the product of experts setup. Intuitively, by training on a set of hard and less biased examples, the weak model learns a stronger explanation for the data which does not rely solely on superficial biases, which characterizes the uncertain/incorrect group, which play a crucial role in debiasing the main model.
>
> | Main model=BERT-base, Weak model=BERT-base |    |    |
> |--|--|--|
> | Weak model's training set                     | MNLI matched (acc.) | HANS Ent (acc.)| HANS Non-Ent (acc.) |
> | Random 2K examples                            |   84.32 | 98.21 |   18.51 |
> | Adv 2K examples (highest loss for hypothesis-only classifier)|   84.41 | 98.85 |   14.42 |
>
> 3. **Though the method in [1] is different, the discussion in that paper still would apply here as well.**
>
> We have made it more clear that our results in Section 5.2 corroborate and complement insights from [1].
>
> [3] Dataset Cartography: Mapping and Diagnosing Datasets with Training Dynamics\
> Swabha Swayamdipta, Roy Schwartz, Nicholas Lourie, Yizhong Wang, Hannaneh Hajishirzi, Noah A. Smith, Yejin Choi\
> EMNLP 2020

---

### Official Review · AnonReviewer4 · 2020-10-28
**Official Blind Review #4**

**Rating:** 7
**Confidence:** 4

**Review:**

## Reason for score

The research problem is critical. The solution is appropriate and novel. The claims are validated. The experiments are interesting.
However, the writing in section 3, 4 et 5 should be improved. If so, I would be willing to raise my score.

## My background

My research is focused on detecting and avoiding data biases (or spurious correlations) learned by deep neural networks. This is the exact scope of this paper. However, my area of expertise is computer vision and multimodal text-image, not natural language processing.

## Summary

Context:
The paper focuses on automatically detecting data biases learned by natural language processing models and overcoming them using a learning strategy.

Problem:
The authors identify and tackle issues of state-of-the-art methods:
- they are required to already know about a certain bias to be overcome.

Solution and novelty:
The proposed method consists in 1) training a weak model that aims at detecting biases 2) overcoming these biases by training a main model using a product of experts (Hinton, 2002) with the predictions of the fixed weak model.

Claim:
- A weak model can be used to discover data biases
- The proposed method produces a main model that generalize better to out-of-distribution examples

## What I liked the most

- meta-problem of automatically detecting and overcoming biases in neural networks is critical
- well contextualized
- relevant issues of state of the art have been identified
- intro and related work are easy to read and understand
- novel, simple and interesting method to tackle them
- interesting figures
- experiments are  interesting and well chosen

## What could be improved

1. Abstract, introduction and 2. Related work
- Your research problem and solution are general and can be applied to many fields. Is there a specific reason why you decided to focus on NLP only?
- You could improve the impact of your approach by citing papers that tackle the same problem with similar solutions from different fields. "Clark et al. 2019 Don’t take the easy way out: Ensemble-based methods for avoiding known dataset biases" that you already cite ran some experiments in multiple fields (NLP, VQA, etc.). "Cadene et al. Rubi: Reducing unimodal biases for visual question answering (NeurIPS2019)" in VQA could also be cited.

3. Proposed Method
- Next to Eq1: Why an element wise sum is equivalent to an element wise multiplication after softmax? It seems wrong to me.
- It could be useful to have a general definition of the PoE loss (instead of just an example of binary cross entropy in Eq2)
- See 4.3, you should define PoE+CE here.

4. Experiments
- Overall, I think it is important that you improve the writing for this section and reduce jargon. It is really difficult to understand for readers that are not familiar with the datasets on which you perform your study. Also it is really difficult to understand which dataset is "in-distribution" or "out-of-distribution".
- You don't define "development matched accuracy" before using it.
4.1
- You use too many footnotes that could be included in the text.
4.2
- You don't define "CE" (even in the caption of Figure2).
- In Table 2, you could reduce jargon by using Weak and Main instead of "W" and "M".
- In Table 2, you don't define "An." even in the caption.
4.3
- I don't understand why "PoE+CE" is better on "Hard"
- I don't like that you propose to use "PoE+CE" as your method of choice "to counteract these effects" without defining it in section 3. To be clear, I still don't understand what is the learning method that you propose PoE or PoE+CE?

5. Analysis
5.2
- Title is on two lines instead of one
- I don't understand "When trained jointly with the larger MediumBERT weak learner......" How many parameters? Don't expect your reader to look at Figure 4 to obtain this information.

6. Conclusion
- Could you add a discussion about the limitations of your approach. In particular: How to choose the number of parameters of your weak learner? What to choose between PoE and PoE+CE? And most critically, if you don't assess the type of biases and the amount of biases included in the dataset, how to be sure that your method will have a beneficial impact? Then, if you need to assess the type of biases, using another method that specifically targets them could be more efficient.

---

> ### Author Response · Authors · 2020-11-24
> **Response to Reviewer4 (1/2)**
>
> We want to sincerely thank you for taking the time to carefully read our submission and providing such detailed suggestions. We have updated our submission by addressing your comments.
>
> **Is there a specific reason why you decided to focus on NLP only?**
>
> Our group expertise is in NLP, and it is an area where these biases have been of particular concern. In theory our method is general, but we make no claims to its empirical impact in other domains.
>
> **You could improve the impact of your approach by citing papers that tackle the same problem with similar solutions from different fields.**
>
> This is a good point. We will add a reference to other works, particularly Cadene et al [2019].
>
> **Next to Eq1: Why an element wise sum is equivalent to an element wise multiplication after softmax? It seems wrong to me.**
>
> We apologize for the typo. The corrected equation is $softmax(e) \propto softmax(w) \odot softmax(m)$     you have noted.
>
> **How to choose the number of parameters of your weak learner?**
>
> We took the smallest BERT model publicly available (TinyBERT with 4.4 million parameters) and did not tune that aspect of the work. Our experiments show that the weaker the pretrained model is, the more common it is to produce “certain but incorrect” responses (Section 5.2). This category gives most of the signals for debiased training.
>
> Other works [Clark et al., 2019; He et al., 2019] suggest that shallow classifiers on top FastText/Glove representations may also lead to good results.
>
> **I still don't understand what is the learning method that you propose PoE or PoE+CE? What to choose between PoE and PoE+CE?**
>
> We found that PoE+CE loss controls the balance between the features from the dataset (superficial cues or not) and the signal from the weak learner. This is similar to how in Distillation it is common to use a mixture of Distill+CE. This indicates that not all the information picked up by the weak model should be discarded.
>
> **And most critically, if you don't assess the type of biases and the amount of biases included in the dataset, how to be sure that your method will have a beneficial impact? Then, if you need to assess the type of biases, using another method that specifically targets them could be more efficient.**
>
> *Start - Copying part of the response to Reviewer3*
>
> This is an interesting point. While it is difficult to enumerate all sources of bias, we focus on “superficial cues” that correlate with the label in the training set but do not transfer. These superficial cues are sufficiently apparent to be captured by a shallow neural network. In a sense, the biases we are targeting are defined by the weakness of the model. For instance, Conneau et al., [2018] suggest that word presence can be detected with very shallow networks (linear classifier on top of FastText bag of words): Table 2 shows very high accuracy for “Word Content”, the probing task of detecting which of the 1’000 target words is present in a given sentence.
>
> To verify that a weak model is still effective with “hard to detect” biases, we consider an example where the bias is only present in a small portion of the training (while the rest of the examples either contradict the bias or are neutral towards this bias). This remaining bias is difficult to spot but should be able to be captured by the weak model. We remove from the MNLI training set all the examples that exhibit one of the two biases detailed in Section 4.1 (high word overlap between premise and hypothesis & entailment; and negation in the hypothesis & contradiction). We end up with 268K examples which present a “low-amount” of bias.
>
> We apply our debiasing method with these 268K examples as our training set. For comparison, we train a main model with standard cross-entropy on a set of 268K randomly selected examples. Our results confirm on HANS that our debiasing method is still effective even when the bias is hard to detect.
>
>
> | Training data | Main Model | Weak Model | Loss | MNLI matched (acc.) | HANS Ent (acc.) | HANS Non-Ent (acc.) |
> | --- | --- | --- | --- | --- | --- | --- |
> | Adversarially selected 268K examples | BERT-base | TinyBERT | PoE  |    83.3 | 92.41 |   38.74 |
> | Randomly selected 268K examples      | BERT-base | ∅               |CE   |   84.01 | 98.15 |    16.50 |
>
> *End - Copying part of the response to Reviewer3*

---

> > ### Author Response · Authors · 2020-11-24
> > **Response to Reviewer4 (2/2)**
> >
> > Regarding the amount of bias, our intuition is the following: in the worst-case scenario, no bias is picked up by the weak learner (it can happen for instance when the bias is present in a tiny fraction of the data). However, Swayamdipta et al. [2020] shows on a range of different dataset that the category “hard-to-learn” examples is always populated. In Appendix A6, we detail the connection between the data maps and our categories and highlight that the “hard-to-learn” examples strongly overlap with our “certain/incorrect” category. It means that even though the weak learner is not picking up biases, there are still examples in the “certain/incorrect” examples which correspond to the hard examples which are upsampled in our PoE training.
> >
> > [Swayamdipta et al., 2020]\
> > Dataset Cartography: Mapping and Diagnosing Datasets with Training Dynamics\
> > Swabha Swayamdipta, Roy Schwartz, Nicholas Lourie, Yizhong Wang, Hannaneh Hajishirzi, Noah A. Smith, Yejin Choi\
> > EMNLP 2020

---

### Official Review · AnonReviewer3 · 2020-10-29
**Straightforward method for reducing model's reliance on spurious features**

**Rating:** 7
**Confidence:** 4

**Review:**

Summary:

This paper focuses on the known problem that current NLP models tend to solve tasks by exploiting superficial properties of the training data that do not generalize. For example, in the NLI task, models learn that negation words are indicative of the label "contradiction" and high word overlap is indicative of the label "entailment". There have been many recent solutions proposed for mitigating such behavior, but existing methods have tended to assume knowledge of the specific dataset biases a priori. In this paper, the authors propose a method based on product of experts that doesn't assume particular knowledge of specific dataset biases. The method works by first training a weak model and then training a "main" model using a loss that upweights examples on which the weak model performs poorly (namely, predicts the wrong answer with high confidence). The assumption is that weak models will exploit heuristics, and so this method will deincentivize the main model to use those same heuristics. The authors evaluate on a range of tasks, including a simulated bias setting, and NLI setting, and a QA setting, and offer a fair amount of analysis of their results. In particular, the analysis showing that the weak learners do in fact adopt the biases which have been documented elsewhere in the literature is interesting, and the discussion of "how weak does the weak learner need to be" is appreciated (a few questions on this below).

Strengths:
* Straightforward method for addressing an important known problem with neural NLP models
* Thorough analysis, not just a "method and results" paper

Weaknesses:
* Novelty might be somewhat limited, method is not wildly creative (but I don't necessarily think "wild creativity" is a prerequisite for scientific value). The authors do a good job of directly contending with the similar contemporaneous work in their paper

Additional Comments/Questions:

Just a few thoughts that came up while reading...
* The weakness-of-weak-learner analysis is interesting. I imagine this is not something that can be understood in absolute terms, i.e., I would not expect there to be some level of weakness that is sufficient for all biases and all datasets. E.g., surely the lexical overlap bias is "harder" to learn than a lexical bias like the presence of negation words, since recognizing lexical overlap presupposes recognizing lexical identity. Therefore, I'd imagine knowing how weak the weak learner needs to be requires some intuition about which biases you are trying to remove, which runs counter to the primary thrust of the paper, namely, removing bias without knowing what the bias is. Thoughts?
* Its interesting that even with this the performance on hans non-entailed is still only 56%, which is better but still not exactly good, and doesn't suggest the model has learned the "right" thing so much as its has learned not to use that particular wrong thing. For research questions such as this ("is the model using the heuristic?") I always find it unsatisfying to think about performance gains that are in between 0 and 100. E.g., when we talk about human learning, we usually see an abrupt shift when the learner "gets it", and our hope in removing the spurious features with methods like yours would be that we'd help the neural models similarly "get it" and reach 100% at least on examples that isolate the effect of this spurious feature. I don't expect you to have an answer for this, but just raising to hear your thoughts.

---

> ### Author Response · Authors · 2020-11-24
> **Response to Reviewer3**
>
> We thank you for your sincere comments.
>
> **I'd imagine knowing how weak the weak learner needs to be requires some intuition about which biases you are trying to remove**
>
> This is an interesting point. While it is difficult to enumerate all sources of bias, we focus on “superficial cues” that correlate with the label in the training set but do not transfer. These superficial cues are sufficiently apparent to be captured by a shallow neural network. In a sense, the biases we are targeting are defined by the weakness of the model. For instance, Conneau et al., [2018] suggest that word presence can be detected with very shallow networks (linear classifier on top of FastText bag of words): Table 2 shows very high accuracy for “Word Content”, the probing task of detecting which of the 1’000 target words is present in a given sentence.
>
> To verify that a weak model is still effective with “hard to detect” biases, we consider an example where the bias is only present in a small portion of the training (while the rest of the examples either contradict the bias or are neutral towards this bias). This remaining bias is difficult to spot but should be able to be captured by the weak model. We remove from the MNLI training set all the examples that exhibit one of the two biases detailed in Section 4.1 (high word overlap between premise and hypothesis & entailment; and negation in the hypothesis & contradiction). We end up with 268K examples which present a “low-amount” of bias.
> We apply our debiasing method with these 268K examples as our training set. For comparison, we train a main model with standard cross-entropy on a set of 268K randomly selected examples. Our results confirm on HANS that our debiasing method is still effective even when the bias is hard to detect.
>
> | Training data | Main Model | Weak Model | Loss | MNLI matched (acc.) | HANS Ent (acc.) | HANS Non-Ent (acc.) |
> | --- | --- | --- | --- | --- | --- | --- |
> | Adversarially selected 268K examples | BERT-base | TinyBERT | PoE  |    83.3 | 92.41 |   38.74 |
> | Randomly selected 268K examples      | BERT-base | ∅               |CE   |   84.01 | 98.15 |    16.50 |
>
> **For research questions such as this ("is the model using the heuristic?") I always find it unsatisfying to think about performance gains that are in between 0 and 100.**
>
> That is a fair point.
>
> For context, McCoy et al [2019] show that the model can “get it” by simply augmenting the training set with heuristic-non-entailed examples (i.e. examples similar to the HANS non-entailed part). With standard fine-tuning, the model gets an almost-perfect accuracy on the HANS non-entailed examples. So we know that for the fixed capacity of BERT-base, we can get to a better minimum (i.e. a minimum that leads to better generalization). Therefore, the question is whether we can optimize the model to reach this minimum without the extra curated data. Intuitively, even though we do not have formal evidence, we can expect that an 80% accuracy reflects the fact the optimization ended in a better generalizing state than when it is at 20% accuracy. Thus, thinking about the 0-100 range as a continuous scale can be helpful (at least maybe outside the ~40-60 range).
>
> [Conneau et al., 2018]\
> What you can cram into a single vector: Probing sentence embeddings for linguistic properties\
> Alexis Conneau, German Kruszewski, Guillaume Lample, Loïc Barrault, Marco Baroni\
> ACL 2018
>
> [McCoy et al, 2019]\
> Right for the Wrong Reasons: Diagnosing Syntactic Heuristics in Natural Language Inference\
> Tom McCoy, Ellie Pavlick, Tal Linzen\
> ACL 2019

---

### Official Review · AnonReviewer5 · 2020-11-06
**Potentially useful extension of prior work**

**Rating:** 6
**Confidence:** 4

**Review:**

*Summary*: This paper proposes a method for training model that are robust to spurious correlations, building upon prior work that uses product-of-experts and a model explicitly trained on a dataset bias (e.g., a hypothesis-only model). Instead of using a model explicitly trained to learn the dataset bias, the authors use a “weak learner” with limited capacity. Then, this model is used in the PoE setting as in past work. The advantage of this method is that a model developer doesn’t need to know that a bias exists, since the hope is that the weak learner will implicitly learn the bias.

*Strengths*: A thorough study of using a limited-capacity auxiliary model to train more robust models, which helps a final model ignore spurious correlations that are easy to learn.

*Weaknesses*: The work is a rather straightforward extension of prior work. Furthermore, the authors only evaluate on 2 textual tasks---I would have liked to see more experiments with spurious correlations in vision (e.g., VQA or the datasets used in https://openreview.net/forum?id=ryxGuJrFvS), and other experiments on text (e.g., the TriviaQA-CP dataset in the Clark paper). As is, it’s hard to glean how broadly applicable this method actually is. I would have also liked to see more of a comparison with methods that use known bias (e.g., Clark et al or He et al)---it seems like some of the comparisons in the table aren’t completely fair.

*Recommendation*: 6 . I think this paper is a potentially-useful extension of a prior method, but I’m still somewhat unconvinced that this method is applicable in settings where the bias is hard to detect, which is what we really care about (since, if the bias is easy to detect, we can use Clark et al and other methods).

Comments and Questions:

1. The comparisons to Clark et al aren’t fair comparisons for adversarial SQuAD, since the Clark et al paper uses a different base model for adversarial SQuAD (modifed BIDAF).

2. The weak learner is a rather blunt instrument. It picks up dataset biases, but it also likely picks up features that are actually useful---not all robust features have to be difficult to learn. Is it possible to better quantify what useful information is being learned (and subsequently thrown out) by the weak learner? This would make it easier to determine if using it is worthwhile.

3. While it’s true that the weak model empirically learns to re-learn the same dataset biases targeted in prior work (e.g., negation correlates with contradiction), it’s somewhat unclear to me how well this method would translate to a setting with unknown biases. The MNLI / SQuAD examples are a bit artificial since we already have knowledge of the bias---it’s possible that weak learners can pick up on spurious features that are “easy to learn”, which are the same ones that humans notice. I’d like to see whether this method applies well to tasks where it isn’t immediately obvious that the bias is easy to learn; perhaps a synthetic experiment would be useful here. Is it possible to modulate the learnability of the bias? The synthetic experiments in the paper suggest that for cases the bias is hard to learn, this method isn’t very effective, which makes sense---in how many of the cases in the literature is the bias hard to learn? This is another reason why I think more experiments would be useful.

---

> ### Author Response · Authors · 2020-11-24
> **Response to to Reviewer5 (1/2)**
>
> 1. **Comparison to [Clark et al, 2019] - BiDAF**
>
> The reported results from Clark et al, [2019] do use a modified BiDAF model while we use a BERT model. Our goal in Table 3 was to highlight the drop of performance when evaluating on the adversarial sets as opposed to in-domain sets. Results from Clark et al [2019] give us a different comparative trade-off in adversarial set performance for a near-state-of-the-art class of model.
>
> 2. **Is it possible to better quantify what useful information is being learned (and subsequently thrown out) by the weak learner?**
>
> Since in-domain performance of the weak learner is an imperfect proxy for learned information, we use a transfer benchmark (see Appendix A.4) to evaluate the generalization capabilities of the weak model. We train the weak model on SNLI using standard cross-entropy and evaluate the transfer capabilities on a range of other natural language inference datasets. We found the following results:
>
> | Transfer Benchmark | Main Model | Weak Model |
> | --- | --- | --- |
> | Test set           | BERT-base (acc.) |  TinyBERT (acc.) |
> | AddOne             |            86.82 |            72.35 |
> | DPR                |            50.29 |            50.01 |
> | SPR                |            58.27 |            42.89 |
> | FN+                |            54.35 |            48.18 |
> | Scitail            |            69.71 |            59.27 |
> | GLUE               |            55.34 |            48.55 |
>
> These numbers suggest that the weak model (TinyBERT) is able to learn some useful information but greatly lags behind the main model (BERT-base).
>
> In general, our experiments suggest that the less generalizable the weak model, the more often it is  “certain but incorrect”. These misclassifications are most beneficial for training the main model. We, therefore, want a weak learner that focuses on less useful information.
>
> 3. **it’s somewhat unclear to me how well this method would translate to a setting with unknown biases [...] I’d like to see whether this method applies well to tasks where it isn’t immediately obvious that the bias is easy to learn**
>
> *Start - Copying part of the response to Reviewer3*
>
> This is an interesting point. While it is difficult to enumerate all sources of bias, we focus on “superficial cues” that correlate with the label in the training set but do not transfer. These superficial cues are sufficiently apparent to be captured by a shallow neural network. In a sense, the biases we are targeting are defined by the weakness of the model. For instance, Conneau et al., [2018] suggest that word presence can be detected with very shallow networks (linear classifier on top of FastText bag of words): Table 2 shows very high accuracy for “Word Content”, the probing task of detecting which of the 1’000 target words is present in a given sentence.
>
> To verify that a weak model is still effective with “hard to detect” biases, we consider an example where the bias is only present in a small portion of the training (while the rest of the examples either contradict the bias or are neutral towards this bias). This remaining bias is difficult to spot but should be able to be captured by the weak model. We remove from the MNLI training set all the examples that exhibit one of the two biases detailed in Section 4.1 (high word overlap between premise and hypothesis & entailment; and negation in the hypothesis & contradiction). We end up with 268K examples which present a “low-amount” of bias.
>
> We apply our debiasing method with these 268K examples as our training set. For comparison, we train a main model with standard cross-entropy on a set of 268K randomly selected examples. Our results confirm on HANS that our debiasing method is still effective even when the bias is hard to detect.
>
>
> | Training data | Main Model | Weak Model | Loss | MNLI matched (acc.) | HANS Ent (acc.) | HANS Non-Ent (acc.) |
> | --- | --- | --- | --- | --- | --- | --- |
> | Adversarially selected 268K examples | BERT-base | TinyBERT | PoE  |    83.3 | 92.41 |   38.74 |
> | Randomly selected 268K examples      | BERT-base | ∅               |CE   |   84.01 | 98.15 |    16.50 |
>
> *End - Copying part of the response to Reviewer3*

---

> > ### Author Response · Authors · 2020-11-24
> > **Response to Reviewer5 (2/2)**
> >
> >
> > 4. **I think more experiments would be useful**
> >
> > Additionally, we have added experiments on one other textual dataset: FEVER [Thorne et al, 2018] using the symmetric challenge set [Schuster et al, 2019]. Our method is again effective at removing potential biases present in training sets. The results are:
> >
> > | FEVER (avg on 6 seeds) |                        | Loss     | Dev set (acc.)    | Symmetric Test set (acc.) |
> > |------------------------|------------------------|----------|-------------------|---------------------------|
> > | Reported               | Schuster et al. (2019) | CE       | **85.85** +/- 0.5 | 57.46 +/- 1.6             |
> > |                        | Mahabadi et al (2020)  | CE       | 85.99             | 56.49                     |
> > |                        | Mahabadi et al (2020)  | PoE      | 84.46             | **66.25**                 |
> > |                        |                        |          |                   |                           |
> > | Ours                   | Bert-base-uncased      | CE       | 85.61 +/- 0.3     | 55.13 +/- 1.5             |
> > |                        | TinyBERT - W           | CE       | 69.43 +/- 0.2     | 43.10 +/- 0.2             |
> > |                        |                        |          |                   |                           |
> > | Ours                   | Bert-base-uncased - M  | PoE      | 81.97 +/- 0.5     | **59.95** +/- 3.3         |
> > |                        | Bert-base-uncased - M  | PoE + CE | **85.29** +/- 0.6 | 57.86 +/- 1.4             |
> >
> > [Thorne et al, 2018]\
> > FEVER: a large-scale dataset for Fact Extraction and VERification]\
> > James Thorne, Andreas Vlachos, Christos Christodoulopoulos, Arpit Mittal\
> > NAACL 2018
> >
> > [Schuster et al, 2019]\
> > Towards Debiasing Fact Verification Models\
> > Tal Schuster, Darsh J Shah, Yun Jie Serene Yeo, Daniel Filizzola, Enrico Santus, Regina Barzilay\
> > EMNLP 2019

---

### Decision · Program_Chairs · 2021-01-07
**Final Decision**

**Decision:**

Accept (Poster)

**Comment:**

This paper considers the problem of learning models for NLP tasks that are less reliant on artifacts and other dataset-specific features that are unlikely to be reliable for new datasets. This is an important problem because these biases limit out-of-distribution generalization. Prior work has considered models that explicitly factor out known biases. This work proposes using an ensemble of weak learners to implicitly identify some of these biases and train a more robust model. The work shows that weak learners can capture some of the same biases that humans identify, and that the resulting trained model is significantly more robust on adversarially designed challenge tasks while sacrificing little accuracy on the test sets of the original data sets.

The paper's method is useful, straightforward, and intuitively appealing. The experiments are generally well conducted. Some of the reviewers raised questions about evaluating on tasks with unknown biases. The authors addressed these concerns in discussion and we encourage them to include this in the final version of the paper using the additional page.